# Effect of Biomass Water Dynamics in Cosmic-Ray Neutron Sensor Observations: A Long-Term Analysis of Maize–Soybean Rotation in Nebraska

**DOI:** 10.3390/s24134094

**Published:** 2024-06-24

**Authors:** Tanessa C. Morris, Trenton E. Franz, Sophia M. Becker, Andrew E. Suyker

**Affiliations:** School of Natural Resources, University of Nebraska-Lincoln, Lincoln, NE 68503, USA; trenton.franz@unl.edu (T.E.F.); sbecker14@huskers.unl.edu (S.M.B.); asuyker1@unl.edu (A.E.S.)

**Keywords:** soil water content, cosmic-ray neutron sensor, biomass correction

## Abstract

Precise soil water content (SWC) measurement is crucial for effective water resource management. This study utilizes the Cosmic-Ray Neutron Sensor (CRNS) for area-averaged SWC measurements, emphasizing the need to consider all hydrogen sources, including time-variable plant biomass and water content. Near Mead, Nebraska, three field sites (CSP1, CSP2, and CSP3) growing a maize–soybean rotation were monitored for 5 (CSP1 and CSP2) and 13 (CSP3) years. Data collection included destructive biomass water equivalent (*BWE*) biweekly sampling, epithermal neutron counts, atmospheric meteorological variables, and point-scale SWC from a sparse time domain reflectometry (TDR) network (four locations and five depths). In 2023, dense gravimetric SWC surveys were collected eight (CSP1 and CSP2) and nine (CSP3) times over the growing season (April to October). The *N*_0_ parameter exhibited a linear relationship with *BWE*, suggesting that a straightforward vegetation correction factor may be suitable (*f_b_*). Results from the 2023 gravimetric surveys and long-term TDR data indicated a neutron count rate reduction of about 1% for every 1 kg m^−2^ (or mm of water) increase in *BWE*. This reduction factor aligns with existing shorter-term row crop studies but nearly doubles the value previously reported for forests. This long-term study contributes insights into the vegetation correction factor for CRNS, helping resolve a long-standing issue within the CRNS community.

## 1. Introduction

The global population is anticipated to reach 9 billion people by the year 2050 [1], which is nearly 1 billion more than the current population in 2024. The growing population will continue to add pressure on land and water resources to meet food production demands. According to recent studies, food production may need to increase at least 56% by 2050 [2]. To meet the additional demand, there will need to be nearly a doubling of current grain production quantities [3]. Agriculture is the leading industrial water consumer [4], withdrawing almost 70% of global freshwater used industrially [3]. Agricultural water use is less than fifty percent efficient, and future demand will likely need double the current agriculturally productive land area [5,6]. These factors make agricultural water conservation a prominent issue impacting current and future generations.

To conserve water in an agricultural context, it is imperative to know how much water needs to be applied to fields (see [7,8]). Water must be saved, but producers must ensure crop vitality and maintain or increase grain yields. One effective way to balance or optimize high yield with maximizing efficient water use is to continuously monitor soil water content (SWC). A technique for quantifying SWC at the scale needed for large-scale sprinkler irrigation (i.e., 100 s of m) is the Cosmic-Ray Neutron Sensor (CRNS). The CRNS is a passive, non-invasive sensor that utilizes epithermal neutron intensity to quantify SWC [9]. The CRNS horizontal footprint is on the order of 10 s of hectares, and the vertical footprint is a depth of decimeters [9,10,11]. Other key advantages are low power demands that can be met by modest sized solar panels; easy data transmission; and measurements that are independent of soil salinity, texture, or bulk density [12].

The CRNS is primarily used in research settings, with nearly 300 sensors globally [13]. Key barriers to its use in agriculture are practical, relating to the high upfront capital cost and the vertical sensing depth not covering the entire root zone for certain crops [14]. As the local price of water (i.e., direct or indirect cost) rises with increased demand, CRNS technology will likely become a more viable option for large-scale irrigation management. With respect to technical aspects, the processing of neutron counts to useable SWC has made excellent progress over the last two decades [10,11,15,16,17,18]. CRNS-moderated detectors capture epithermal neutrons [13,18,19], where the neutrons are primarily slowed down by interactions with hydrogen at this energy level [9,15]. In nearly all ecosystems, hydrogen within the CRNS footprint is dominated by water in the soil (60–90%) [20]. However, water vapor in the air [21] and liquid water in plant biomass may also account for significant amounts of the ecosystem’s hydrogen (10–40%) [22]. These variations can have significant impacts on SWC estimates.

Figure 1 shows the impact of variations in the neutron count calibration parameter (*N*_0_, see Equation (2) in Section 2.3 for more details) on SWC estimates. Systematic errors in SWC can manifest due to neglecting the various correction factors on the raw neutron counts. CRNS processing has developed several epithermal neutron correction factors to isolate the SWC signal in the soil. This includes first- and second-order scale correction factors for time-varying air pressure (1st), high-energy neutron intensity (1st), and atmospheric water vapor (2nd), as summarized in Zreda et al. [15] and continually improved by the community [23]. However, a universally accepted neutron correction factor for variations in vegetation biomass and types (i.e., crops vs. grasslands vs. forests) remains an unresolved issue.

Numerous studies report on vegetation’s influence on CRNS measurements (Table 1). The most widely cited vegetation correction factor was proposed by Baatz et al. [24] regarding biomass (*f_b_*) for Norway spruce trees. The study found a 0.9% linear reduction in epithermal neutron intensity for every 1 kg m^−2^ of dry above-ground biomass. The factor was extrapolated to 0.5% neutron count reduction for every 1 kg m^−2^ of biomass water equivalent (*BWE*). Vather et al. [25] reported a 13.8% reduction in *N*_0_ when 13.7 kg m^−2^ (dry above-ground biomass) of acacia trees were removed for timber production, where *N*_0_ is a free calibration parameter in Desilets et al.’s [18] calibration equation for converting neutron counts to SWC. Franz et al. [22,26] also found a linear reduction with *N*_0_ of about 1% per unit *BWE* (kg m^−2^) for maize and soybeans. The main limitation of these studies is that they were short-term or used indirect biomass estimates in forest ecosystems, where direct sampling is not practical. To the authors’ knowledge, no existing long-term studies have quantified the effects of biomass variations on epithermal neutron intensity. Heistermann et al. [27] did present a 3-year study, but no equation for *f_b_* was reported.

Vegetation measurements can be performed in several ways (Table 1). Relatively few studies [22,26,28,43] have utilized direct biomass water equivalent (*BWE*) measurements, as it requires destructive sampling, which is time- and labor-intensive. *BWE* is defined as the amount of water in biomass tissues [22], which can be expressed as an equivalent depth of water (mm). Rather than treating biomass as above-ground biomass (AGB) or below-ground biomass, *BWE* uses measurements of standing wet biomass and standing dry biomass to determine the water content of the plants. Measurements of *BWE* are practical in row crops but impractical for forests. Alternatively, forestry studies have used allometric relationships [44], like stand density and base diameter, to estimate dry biomass in an area. However, this ignores any internal water content variations within the canopy.

Row crops and forests differ in standing wet biomass, *BWE*, and *f_b_* (see conceptual Figure 2). The vegetation correction factors proposed have primarily been determined in forest environments [24]. However, the growing period of a forest stand is much longer than that of maize or soybeans (Figure 2). This means that while forests eventually have more wet biomass and *BWE* than crops, this trait can take several years to develop, with small inter-annual variations. Conversely, the peaks of crop biomass and *BWE* are smaller than forests, and inter-annual variations are large (Figure 2). Forest wet biomass reaches a large, sustained value, while crops have smaller (nearly 25% the amount in this example), sharp peaks. With respect to internal water content, both row crops and forests follow a seasonal pattern, but forests are presumed to have a much lower amplitude [22,25]. Much of the water stays in the forest year round, as only leaves are lost, and the stem dries slightly. Crops reach senescence in the fall and lose most of their water content prior to harvest (typical harvest for maize and soybeans is at 25% or less). After harvest, the only water content remaining in the crops is in the roots and shoots, which desiccate until the next growing season.

Long-term validation studies of the *f_b_* correction factor in forest and agricultural ecosystems are limited. The lack of an accepted *f_b_* correction factor may cause systematic error in CRNS SWC observations. A long-term study was performed in eastern Nebraska with three field sites in a maize–soybean rotation. The study’s goal was to investigate if the change in epithermal neutron intensity with *BWE* was linear and if the slope was consistent among years and between crops. Additionally, we assessed if a sparse network of time domain reflectometry (TDR) sensors was viable to quantify changes in neutron intensity with *BWE* as opposed to using intensive soil sampling campaigns.

## 2. Materials and Methods

### 2.1. Study Area

Located approximately 10 km south of Mead, Nebraska, USA, the Eastern Nebraska Research, Extension, and Education Center (ENREEC) has over 3500 hectares of research and operational crop and livestock facilities. This study utilized three field locations, identified as CSP1, CSP2, and CSP3 (CSP stands for Carbon Sequestration Project and is the naming format used to identify fields at ENREEC. In the Ameriflux identification system, these fields are known as US-Ne1, US-Ne2, and US-Ne3.). CSP1 and CSP2 have had CRS-2000/B sensors active since 2019, while CSP3 has had a CRS-1000/B active since 2011 (Figure 3). The key differences between the CRS-1000/B and CRS-2000/B are the gas volumes and pressures. The CRS-1000/B has a gas volume of ~18 L and a pressure of 0.6 atm, and the CRS-2000/B has a gas volume of ~36 L and a pressure of 0.6 atm. Due to these factors, the CRS-2000/B counts approximately twice as many neutrons as the CRS-1000/B. Greater count rates lower the uncertainty of measured values. These sensors were initially installed as part of the COsmic-ray Soil Moisture Observing System (COSMOS) project [15]. The soil is primarily clay loams, where water is easily retained with a low hydraulic conductivity [45]. The study fields are also part of the Ameriflux and Long-term Agricultural Research networks [46,47,48].

At site CSP1, maize was grown in 2019, 2020, 2021, and 2023, whereas soybean was grown in 2022. At sites CSP2 and CSP3, maize and soybean were rotated annually, with maize sown in odd years. All three sites were no-till, and all plant residues were left in place for soil cover. CSP1 and CSP2 were irrigated by overhead sprinkler center pivots, and CSP3 was rainfed. The driest year in the study was 2020, with 306 mm of precipitation from 1 April to 15 November, while the wettest year was 2015, with 781 mm of precipitation in the same time frame. Generally, most irrigation to CSP1 and CSP2 occurred in July and August. The largest quantity of irrigation water applied to both CSP1 and CSP2 occurred in 2022, with CSP1 having 299 mm and CSP2 having 251 mm applied. The smallest irrigation quantity for CSP1 was in 2019, with 130 mm applied, and for CSP2 was in 2021, with 88 mm applied.

The yearly study period was generally from 1 April to 15 November, depending on the CRNS installation date and data availability. The period included plant pre-emergence and post-harvest conditions to allow for bare soil neutron count estimates. The CRNSs at CSP1 and CSP2 were installed on 31 May 2019 and at CSP3 was installed on 20 April 2011. In 2023, TDR data were available through 8 November. This analysis includes only the April to November period to minimize the effects of snow on CRNS data and potentially frozen soil conditions, which affect TDR measurements. Data were collected during the winter period but are not included in this analysis.

### 2.2. Description of SWC Datasets

CRNS calibration (see Equation (2)) requires independent measurements of SWC averaged across the footprint [9,15,18]. Each ENREEC field had at least three TDR sensor profiles at multiple depths, which collected hourly SWC. CSP1 and CSP2 each had three TDR sensor profiles, while CSP3 had four. The TDR sensors were installed as part of the Long-Term Agricultural Research network and Ameriflux projects, but they were not for direct use in this study. For this reason, the TDR sensors were installed at depths to meet the goal of those projects, which did not necessarily align with the goals of this study. We consider this a sparse TDR network for use with CRNS data analysis due to the spatial heterogeneity within the ~20 ha CRNS footprint (see [15,49,50] for more discussion). The TDR sensors were installed at depths of 10, 25, 50, and 100 cm for all fields and an additional depth of 175 cm for CSP3 only. For comparison with CRNS, the depths of most influence are 10 and 25 cm. Given the sparse data, we assumed a simple weighting of 75% for the 10 cm depth and 25% for the 25 cm depth in the analysis. The vertical weighted average for each location was then averaged across all locations. The 75% and 25% weights were deemed a reasonable and simplistic estimate to capture the increased influence the upper depth contributes to the neutron signal following Zreda et al. [9], Franz et al. [49], and Schrön et al. [11]. Neutron contribution to the CRNS varied with depth and water content. All fields generally had an SWC of 20% or greater, and with this water content, the CRNS had minimal influence at depths below 25 cm, so these deeper sensors were not included in the depth weighting [9,11,49]. A more rigorous weighting analysis following Schrön et al. [11] will be used for the more comprehensive SWC data from the 2023 gravimetric field campaigns. CSP1 and CSP2 had eight gravimetric sampling campaigns, while CSP3 had nine. The sampling pattern followed that proposed by Schrön et al. [11], with data collected at 19 locations (one at CRNS and 18 at radii of 10, 50, and 125 m every 60 degrees) and every 5 cm down to 30 cm. The gravimetric results were weighted vertically in three ways: simple arithmetic, the TDR weighting using 75% and 25% for the 10 and 25 cm depths, and the weighting method proposed by Schrön et al. [11]. Recently, Schrön et al.’s [11] weighting method has been proposed as the most accurate weighting method, accounting for both radial distance and depth of neutron influence. This study used the 10 and 25 cm weightings to directly compare TDR measurements and arithmetic average weighting to a prior study in this same field [22].

### 2.3. Description of CRNS Data

Hourly raw moderated neutron counts (*N*′) were collected for each CRNS. The raw counts were first filtered to eliminate extraneous values based on expected count rate minimums and maximums for each site. Next, standard correction factors for variations in pressure (*f_p_*), air humidity (*f_v_*), and neutron intensity (*f_i_*) were applied following Zreda et al. [15]:(1)$$N_{pvi}=N^{\prime }*f_{p}*f_{v}*f_{i}$$

Next, a Savitsky–Golay filter was applied to the corrected neutron count data (*N_pvi_*) following Franz et al. [51]. To convert corrected neutron counts to SWC, the modified Desilets et al. [18] equation was used:(2)$$\left(\theta_{p}+\theta_{LW}+\theta_{{SOC}_{eq}}\right)=\frac{0.0808}{\frac{N_{pvi}}{N_{0[f\left(BWE\right)]}}-0.372}-0.115$$
where $\theta_{p}$ is the pore water content (g g^−1^), $\theta_{LW}$ is the lattice water content (g g^−1^), $\theta_{{SOC}_{eq}}$ is the soil organic carbon water content equivalent (g g^−1^), *N_pvi_* is the corrected neutron counts (counts per hour, cph), and *N*_0_ is the instrument-specific count rate for dry silica soil (cph). We note that *N*_0_ is a function of the vegetation biomass water equivalent (*BWE*) present in the detector’s support volume [22,24,26]. Lastly, the volumetric SWC can be found by multiplying the pore water content by $\rho_{s}/\rho_{w}$, where $\rho_{s}$ is the dry soil bulk density (g cm^−3^), and $\rho_{w}$ is the density of water (assumed to be 1 g cm^−3^).

Laboratory studies determined $\theta_{LW}$ and $\theta_{{SOC}_{eq}}$, while $\rho_{b}$ was determined by gravimetric sampling in 2011 and 2023 (Table 2). Both years of gravimetric sampling yielded similar values. $\theta_{LW}$ and $\theta_{{SOC}_{eq}}$ were determined by gathering an ~100 gm aggregate soil sample from each location (~5 gm each) for each field at depths of 0–10 cm, 10–20 cm, and 20–30 cm. The samples from each field were air-dried and sent to Activation Laboratories Ltd. in Canada, where infrared spectroscopy (IR absorption) was utilized to determine $\theta_{{SOC}_{eq}}$, and $\theta_{LW}$ was determined by thermal decomposition in a resistance furnace followed by IR absorption. Given the proximity of the fields and similar results, the average value of the three fields was used in the data processing. The detection limits in determining $\theta_{LW}$ and $\theta_{{SOC}_{eq}}$ were 0.5% and 0.1% by mass, respectively. One item of note was the increase in $\theta_{{SOC}_{eq}}$ between 2014 and 2023. A possible explanation for this was crop litter, which will be further discussed. Another factor that may have influenced this was a change in sampling methodology between 2014 and 2023. In 2014, total carbon and CO_2_ were measured, and the organic carbon was calculated using stoichiometry. In 2023, organic carbon was measured as well.

### 2.4. Biomass Water Equivalent Measurements

At each field site, destructive vegetation measurements were taken approximately every 10–12 days during the growing season (13 years for CSP3, five years for CSP1 and CSP2). These measurements capture changes in biomass throughout the year and over the various growth stages of maize and soybeans. For each sample date and field, three complete plants (above ground only) were removed from 6 intensive management zone (IMZ) locations. These IMZs were 10 m × 10 m areas surrounding the markers in Figure 3. Only 3 or 4 of the 6 IMZs had associated TDR sensors with them, which are shown in Figure 3. The plants were immediately weighed and then dried for five days at 70 °C. This methodology provided standing wet biomass (*SWB*, kg m^−2^) and standing dry biomass (*SDB*, kg m^−2^). *BWE* (kg m^−2^) was then be calculated as:(3)$$BWE=SWB-SDB+SDB*f_{WE}$$
where *f_we_* = 0.494, accounting for the water equivalence in the cellulose [26,40]. Note that *BWE* can be expressed in depth of water (kg m^−2^~mm). This was done to avoid confusion with the biomass measure of above-ground biomass (AGB), which was used to determine a biomass correction factor for CRNS in prior studies. The Appendix A contain figures for biomass growth from all sites and all years. Linear interpolation was used between measurement dates to obtain a daily *BWE* value. The maize and soybean crops follow the same general trend, with low amounts of *BWE* at the beginning of the year as the plants emerge, followed by a rapid increase in biomass until a peak is reached. After this peak, the plants rapidly dry out until harvest at around 25% water content. The *BWE* maximum for soybeans is generally 3–4 mm and 7–8 mm for maize.

At the end of each season, the crop grown that year is harvested. The soil surface is not cleaned, so a small litter layer of plant matter remains on the surface. In the years that maize was grown, there was approximately 50 cm tall stover left over on the ground surface. This stover was very dry, and it was estimated that this stover contained 0.5 mm or less BWE. In the years when soybeans were grown, the plants were harvested very close to the ground, so most of the previous years’ stover was removed as well. Any litter that remained was likely blown away or desiccated over the winter season. Some litter may have remained and become incorporated into the soil as soil organic carbon ($\theta_{{SOC}_{eq}}$), which may account for the slight increase in $\theta_{{SOC}_{eq}}$ in CSP3 between 2014 and 2023 (Table 2).

### 2.5. Statistical Analysis of BWE on Neutron Intensity

Daily *N*_0_ values were calculated using Equation (2) and the weighted TDR and gravimetric datasets. Linear regression was used to find a best-fit slope and intercept for the relationship between *N*_0_ and *BWE* using Python’s NumPy and SciPy packages [52,53]. The coefficient of determination (r^2^) and 95% confidence intervals were calculated for each coefficient. The fitted slope and intercepts were compared between each year and field. As discussed above, three different weighting schemes of the 2023 gravimetric data were used. Due to different CRNS types, the comparison between fields could only be made using the slope-to-intercept ratio. The slope-to-intercept ratio (*η*), also known as the neutron intensity reduction coefficient, was first reported by Baatz et al. [24] and is a useful metric that represents the relative percent decrease in neutron intensity per unit increase in biomass.

Baatz et al. [24] found *η* = 0.5% for *BWE* in a forest site, while Franz et al. [22,26] found *η* = 1% for *BWE* in crops (see Table 1 above). Baatz et al. [24] suggested applying a biomass correction factor (*f_b_*) to Equation (1):(4)$$N_{pvib}=N^{\prime }*f_{p}*f_{v}*f_{i}*f_{b}$$
where
(5)$$f_{b}=\frac{1}{1-\eta *BWE}$$

Here, *f_b_* is the residual information comparing *N_pvi_* vs. *N_estimate_*, where *N_estimate_* can be found using the gravimetric or TDR SWC data, *BWE*, and Equation (2). To calculate *N_estimate_*, an *N*_0_ value is also necessary, which is the average intercept of the *N*_0_ vs. *BWE* relationship. We note this *N*_0_ is where *BWE* = 0 and will be denoted as *N*_0,*BWE*=0_. Figure 4 shows an overall workflow of these calculations.

## 3. Results

### 3.1. Overview of SWC, BWE, and CRNS at Study Sites

Figure 5a presents measured *N_pvi_*, *N*_0_, and *BWE* values for CSP1 in 2023. The plot illustrates that *N_pvi_* and *N*_0_ followed similar general patterns, as peaks have nearly the same timing and shape, and only the amplitude varies, with more subdued peaks for *N_pvi_* values. The amplitude variations exist because the *N*_0_ values account for the pore water content, lattice water content, soil organic carbon, and fixed parameters, while *N_pvi_* does not. This similarity in shape is anticipated, as *N*_0_ is calculated using the modified Desilets et al. equation (Equation (2)) using *N_pvi_* as an input value. Most notably, *N*_0_ was not constant but had a pronounced dip in the summer months, corresponding to the peak *BWE* at the site. This *BWE* impact does not noticeably manifest in *N_pvi_* plots. *N*_0_ is a calibration parameter that includes the effects of *BWE*; other unaccounted hydrogen pools; or potentially over/under neutron intensity correction factor effects due to *f_p_*, *f_i_*, and *f_v_*. Figure 5b illustrates the precipitation, irrigation, and TDR SWC content. As expected, when there was limited precipitation, SWC decreased, as shown by low TDR time series peaks. If precipitation increased or irrigation was applied, SWC increased. When TDR values reached a low point, especially for a sustained period, irrigation was applied to increase SWC values and maintain crop production at this site.

### 3.2. Influence of BWE on Neutron Intensity

Ideally, with all correction factors applied in Equation (1), *N*_0_ should be constant over time. A clear relationship existed between the *N*_0_ and *BWE* time series for CSP1 in 2023 using TDR data and Equation (4) (Figure 6). The relationship between *N*_0_ and *BWE* was generally consistent across all sites and years (Appendix A).

*N*_0_ was not constant in time and contained considerable noise given the limited number of TDR sensors used to estimate *N*_0_ (Figure 6a). There was a shift in June, which was likely due to a sensor error, which was then repaired. In the middle of the growing season (May through August), there was a significant decrease in *N*_0_. After this low point, *N*_0_ increased again from August through November. This time frame for alterations in neutron counts is significant, as from May to August, crops are actively growing and increasing in wet biomass and *BWE*. Biomass starts low as plants are beginning to emerge (Figure 6b). This is followed by a rapid increase in *BWE* as the plant grows, primarily in wet biomass, until senescence begins in August. During senescence, the plants maintain their dry biomass but lose most wet biomass, shown by a decrease in *BWE*. After harvest, *BWE* remains higher than zero to account for the plant debris, like corn stover, on the surface. Most of this debris degrades over the winter and is completely removed with soybean harvests the following year. The low point in the *N*_0_ plot coincides with the highest point in *BWE*. This leads to the inference that increased *BWE* decreased neutron counts and *N*_0_ (Figure 7).

Figure 7 shows a scatter plot and statistical summary of the correlation between *N*_0_ and *BWE*. Additional plots for all fields and years can be found in the Appendix A. This plot shows a clear linear relationship between *N*_0_ and *BWE* using both the gravimetric and TDR data for all fields and all years (Figure 8). 

In addition to using the TDR data, Figure 8 shows the results using the gravimetric data from 2023 and the three different weighting schemes. The key differences between using TDR data and gravimetric data are the accuracy and uncertainty. Gravimetric data are considered the most accurate SWC measurement method, but they are difficult to collect, which is why this study only has eight or nine gravimetric SWC measurements per field. TDR data were collected hourly and then averaged to daily values, meaning one data observation per day. Over the course of the growing season, this was 228 observations. The gravimetric data are considered a more accurate SWC estimate, but the values have more uncertainty due to the small number of observations compared to TDR data. The weighting schemes had some influence on the intercept value (shifting it up and down) but had minimal influence on the slope of gravimetric values. For the slope, all gravimetric slopes fell within each other’s confidence intervals, which is shown below in Table 3. The TDR slope and intercept values varied from the gravimetric values, which is shown below. For CSP1, the highest r^2^ was found for the 2023 Schrön et al. weighted gravimetric data at 0.81, and the lowest was the average of all TDR data at 0.41. For CSP2, the largest r^2^ value was 0.88 for the 2023 arithmetic gravimetric data, and the lowest was 0.38 for the TDR data. Finally, for CSP3, the highest r^2^ value was 0.38 for the arithmetic-weighted gravimetric data, and the lowest was 0.05 for the average of all TDR data. Except for the CSP3 TDR data, all r^2^ values were high (>0.3). The exception was CSP3 using TDR, likely due to a positive slope in one year (2020) and relatively low negative slopes in other years. We also found that year-to-year variation between each field’s *N*_0_ and *BWE* slope and intercept existed (Figure 9).

In general, the slopes and intercepts were consistent, but there was a notable exception for CSP3 in 2020 (Figure 9f). This was likely due to a malfunction in the CRNS sensor. We note the battery was intermittent over the spring/summer period due to COVID-19 restrictions preventing routine maintenance and inspection. The battery was finally able to be replaced in July 2020. We note that CSP1 and CSP2 did not have the same problems for this period. For the remaining analyses, we excluded CSP3 2020 data as an outlier.

The reported slopes represent the change in *N*_0_ with *BWE*. The slopes of CSP1 ranged from −36.22 to −13.28, with an average TDR value of −27.0 and an average of all three 2023 gravimetric weighting methods of −31.1. For CSP2, the slopes ranged from −67.36 to −24.99, with an average TDR value of −38.6 and an average 2023 gravimetric value of −49.7. For CSP3, the slopes ranged from −2.11 to −21.61, with a TDR average of −12.6 and a 2023 gravimetric value of −9.6. Within all three fields, the confidence intervals overlapped between several years, indicating they fall within one another’s error ranges (please see Figure 9 to view confidence intervals). Most TDR slope values within each field were within the 10% error of the average slope value for CSP1 and CSP2. This same consistency likely was not seen in CSP3 due to more variability in measurements. We note that while slope values may not be directly comparable between each field, the similarities can be easily seen in the multiyear plots (Figure 8). Additional information on the slope and intercept values and their statistical relationship between years can be found in the Appendix A.

The intercepts were mostly consistent. The intercept values for CSP1 ranged from 2793.91 cph to 2931.95 cph, with a TDR average of 2890.7 cph and a gravimetric average of 2830 cph. For CSP2, the lowest value was 2864.39 cph, the highest was 3044.36 cph, the TDR average was 2947.3 cph, and the gravimetric average was 2920.86 cph. For CSP3, the lowest value was 1542.13 cph, the highest value was 1688.51 cph, the TDR average was 1614.1 cph, and the gravimetric average was 1582.18 cph. All values were similar between fields CSP1 and CSP2. CSP3 had larger variability due to the lower count rates of the CRS-1000/B sensor. Looking within each field, year-to-year intercept values aligned well. The confidence intervals overlapped for many years presented (Figure 9). All within-field intercept values were within 200 counts, which is within the 5% error range of the average TDR value. For CSP1 and CSP2, there was a significant decrease in intercept values in 2023. For CSP3, there seemed to be a systematic shift to higher intercept values from 2016 to 2023. To the authors’ knowledge, there is no clear physical explanation for this shift. One potential source of this change was an increase in soil organic carbon; however, as previously mentioned, most of this was removed every two years. While this was within the range of the overall average, this area deserves additional investigation. The relative magnitude of the error bars for gravimetrically determined SWC was large due to fewer gravimetric observations collected over the season as compared to the number of TDR SWC measurements (~8/9 for gravimetric and 228 for TDR) (Figure 9).

Table 3 summarizes the ratio of slope to intercept, *η*, which defines the percent change in *N*_0_ with *BWE*. The key finding in Table 3 is that while there were small variations between the slopes and intercepts of gravimetric weighting methods, and there was interannual variability in TDR slopes and intercepts, the value of *η* remained similar within each field. Between CSP3 and the other two fields, there were differences, but within each field, the values were consistent. The minimum value for CSP1 was −1.4, the maximum was −0.45, the average TDR value was −0.9, and the average gravimetric for the three weighting schemes was −1.10. The minimum value for CSP2 was −2.28, the maximum value was −0.82, the average TDR value was −1.3, and the average of the three gravimetric schemes was −1.70. For CSP3, the minimum value was −1.31, the maximum value was −0.13 (excluding 2020), the average TDR value was −0.59, and the average gravimetric value of the three weighting schemes was −0.61. The *η* values for CSP3 were approximately half the *η* values of the other two fields. Again, note that CSP3 had a different detector with half the counting rate of CSP1 and 2, meaning larger uncertainties in the count rates. Also, only five years are available for CSP1 and CSP2 compared to 13 years for CSP3. With small sample sizes, the 95% confidence interval will be larger and subject to extreme values. Most notably, CSP2 in 2022 was lower, thus influencing the overall average. Despite the uncertainty in *η*, it was generally consistent year to year. There were no obvious variations between maize and soybean years.

### 3.3. TDR vs. Gravimetric Sampling

The results presented above illustrate the similarity between gravimetric data and TDR data for detecting the influence of *BWE* on neutron intensity at the seasonal timescale. With an intense year of gravimetric data collection and TDR data, this study provides a unique opportunity to compare two common SWC estimation techniques. A direct comparison of 2023 gravimetric data by various weighting methods with 2023 TDR data was performed (Figure 10).

Figure 10 illustrates the variation in the slope and intercept using the TDR and three different weighting methods for the gravimetric data. The arithmetic average method assumes that all points in the sampling radius have equal contribution to the CRNS signal. Based on knowledge of CRNS footprints, points further away and deeper samples will have a lower contribution to the detector. The Schrön et al. weighting method is the most comprehensive, but we did not find substantial improvement in the fitting of *N*_0_ vs. *BWE*. This figure reflects the same variations in slope and intercept as seen in Figure 9. The intercept shifted upward or downward depending on the gravimetric weighting strategy employed, with the gravimetric and 10 and 25 cm weighting having very similar intercepts. The slope was consistently negative, with all gravimetric confidence intervals overlapping (see Table 3). One possible explanation for differences between TDR and gravimetric values is that the gravimetric data captured the upper soil depths (above 10 cm), while the TDR sensors did not capture this range, where SWC is the most variable.

### 3.4. Proposed Biomass Correction Factor, f_b_

Per the data shown above, there was a linear relationship between *N*_0_ and *BWE*, allowing for a simple correction factor, *f_b_*, to be proposed. Using Equation (2), we can solve for *N_pvi_*, which we refer to as *N_estimate_*. This required using an SWC dataset (either from TDR or gravimetric sampling) and an estimate of *N*_0_. For *N*_0_, we used the best-fit intercept values (i.e., *N*_0,*BWE*=0_) provided in Table 3. The lattice water and soil organic carbon values remained the same for all sites at 0.058 m^3^ m^−3^ and 0.0092 m^3^ m^−3^, respectively. The *N_estimate_* values were then compared to the detector-measured *N_pvi_* values. Differences in the *N_pvi_* vs. *N_estimate_* were most influenced by additional hydrogen pools in the CRNS support volume, here being the crop *BWE*. The residual values between *N_pvi_* and *N_estimate_* (*f_b_*) values were plotted against *BWE* (Figure 11) along with Baatz et al.’s [24] correction factor. Baatz et al. [24] proposed a *BWE* vegetation correction factor (*f_b_*) of 0.5% neutron reduction per kg of *BWE*.

The data presented here illustrate that the slope of the line is positive, corresponding to Baatz et al., but with a larger magnitude than previously found (Figure 11). Table 4 contains a summary of all weighting schemes for each field. CSP1 and CSP2 showed larger slopes for the gravimetric data as compared to the TDR values. CSP3 showed similar results between the two, but with much greater uncertainty, as indicated by the reported r^2^ values. We also note that the slope values for CSP3 were more in line with Baatz et al. (0.005), whereas CSP1 and 2 were two to three times greater. One potential reason for the agreement in CSP3 and the discrepancy between CSP1 and CSP2 could be sensor type. Baatz et al.’s [24] study utilized a CRS-1000/B CRNS, similar to CSP3.

Table 4 indicates that regardless of the gravimetric weighing method used, the *f_b_* determined here is larger than the original Baatz et al. correction factor. We note that the Schrön et al. weighting method performed the worst. One explanation is that Baatz et al.’s factor was released in 2015, two years before the Schrön et al. weighting method was published. The TDR, arithmetic, and 10 cm and 25 cm weighting of gravimetric samples behaved similarly for *f_b_*. This is interesting, as the arithmetic average included samples from the upper 10 cm of soil, while the 10 cm weighting, 25 cm weighting, and TDR did not. This upper layer of soil (0–10 cm) contributed the most signal to the neutron count and experienced the most variation in SWC. However, the lack of samples in the top 10 cm did not greatly impact the overall fits reported in Table 4.

## 4. Discussion

### 4.1. Relationship between N_0_ and BWE

By plotting *N*_0_ vs. *BWE* (Figure 7 and Figure 8), it is clear that a linear relationship exists between these variables, affirming the findings by Franz et al. [26], Tian et al. [28], Jakobi et al. [29], Heistermann et al. [27], and Jakobi et al. [43]. The data presented in Figure 8 indicate that *N*_0_ can be accurately represented by variations in *BWE* using a linear model, and a simple *f_b_* correction factor can be used for row crops. Figure 9 also shows that there was considerable noise and year-to-year variability, but there was consistency in estimated slope and intercept values. The slope values for CSP1 and CSP2 were very similar, while CSP3 differed slightly. One reason for this difference may be the different detector used in CSP3. The CRS-1000/B counted half as many neutrons as the CRS-2000/B, resulting in larger neutron count uncertainty. The lower count rates blurred the relationship between *N*_0_ and *BWE*, as reflected in the lower r^2^ values. Ideally, the intercept of the fit should not change year to year if all neutron correction factors and hydrogen pools are accounted for properly. In prior studies, only a maximum of two years of data were reported, which did not allow for a comprehensive understanding of how the intercept and slope values may change from year to year. Given the consistency in results year to year, this indicates the same correction factor can be used for soybeans and maize and likely other crops with similar root-to-shoot ratios. However, we note that for crops that grow primarily below ground (i.e., potatoes, yams, cassava), additional studies or an updated sampling methodology may be needed to determine the *f_b_* correction factor.

### 4.2. Sparse TDR Network vs. Gravimetric Sampling

The best r^2^ fit among the weighting methods and two types of SWC data changed between each field (Figure 10). Ideally, one weighting method or sampling methodology would present the highest coefficient of determination. This was not the case, as in CSP1, the Schrön et al. weighted method best defined the relationship; in CSP2, the arithmetic average best defined the relationship; and for CSP3, TDR best defined the relationship. This result may have been impacted by when the samples were collected. For example, if sampling occurred the day after irrigation or precipitation in one field versus another, or if more days with high biomass samples were taken in the gravimetric sampling in one field versus another, this may have altered relationships.

Due to the spatial locations and depths selected, the sparse TDR network likely contains significant bias for representing the entire field and CRNS footprint. The TDR sensors were selected based on the Ameriflux experiment and research objectives, not to validate the CRNS methodology. As shown in Figure 3, many of the TDR sensors were near one another within the field, which limited the spatial variability captured by the sensors. The lack of spatial coverage within the field may have skewed the results to the measured SWC conditions near the sensors and ignored the remainder of the field. However, the most notable bias stemmed from the TDR sensors placed at 10 cm and 25 cm depths. Using these depths meant that any wetting events, like irrigation or precipitation, occurring in the top 10 cm may not have made it deep enough to be detected by the TDR sensor. It is possible that much of the water in the upper layer would be intercepted by plant roots or be involved in the evapotranspiration process. Variability in the top 10 cm would also be much greater than the variability at deeper sensors.

With respect to CSP1 and CSP2, we found the TDR slope value was less steep than the gravimetric slope values, while in CSP3, the TDR relationship was steeper. The timing of the gravimetric sampling likely caused some of these differences. In CSP1 and CSP2, two SWC data clouds were present, with a large gap in the middle *BWE* range. There were sample points when there was a small *BWE* present, and there were points taken when there was >5 mm *BWE*. For CSP3, the data were more continuous, with points in the middle of the *BWE* spectrum. The data clouds resulted from the maize growth pattern and sampling week pattern. The sampling dates operated in the following fashion: week 1—CSP3, week 2—CSP1 and CSP2, week 3—CSP3, week 4—CSP1 and CSP2, and so on. However, we note that sampling was affected in late June due to the application of fertilizers and pesticides through the irrigation system, which did not allow entry into the fields for 1–2 days. Maize grows rapidly during certain growth stages (from mid-June to mid-July). Initially, the biomass increase is slow (May to June), followed by a rapid increase in growth. From the destructive biomass samples collected, the growth stages progressed exceptionally fast, starting in mid-June (from stage V8 to stage R6 over 97 days). The maize grows at such a rate that this whole range of biomass is missed by missing one week of sampling. This same effect was not as severe in the CSP3, as it was rainfed with slower growth rates. CSP1 and CSP2 also applied chemigation and fertigation, encouraging growth and minimizing hindrances to maize progression. This growth rate and missing every other week of sampling means many growth stages were missed in the fast-growing fields, while these stages can be seen in CSP3. Due to these reasons, two clusters of points in CSP1 and CSP2 existed, resulting in the statistical fitting being influenced by the low and high biomass values more than CSP3, which had better consistency between *BWE* values. Overall, the linear relationship was still strong for CSP1 and 2. The use of the TDR sensors and remote sensing could help fill in these sampling gaps.

Another factor altering the gravimetric values compared to the TDR values was the number of data points. There were only eight gravimetric points for CSP1 and CSP2 and nine for CSP3, with many more samples for the TDR network (>200). This means that every point in the gravimetric data had substantially more influence on the slope of the line and the r^2^ of the relationship. For example, in CSP1, there was a very high point, making the line steep. For CSP2, there were very high and very low points on the plot, making the slope of the line much steeper. In CSP3, there were no severe high or severe low points, and there were middle points, making the slope of the line flatter. The TDR data slope relies on many points, and one extreme high or low point will not influence the line greatly.

Another goal of this study was to determine if a sparse TDR network can be used as a reliable measurement of pore water content in the *N*_0_ equation (Equation (2)) in determining the influence of *BWE*. At each site, there was only a maximum of four TDR locations with five depths, whereas the gravimetric data better captured field variability. From our results, we found that TDR did provide reliable data for seasonal averages. Despite some differences in the slope and intercept, overall, the results were consistent, as shown in Table 4. Taken together, the gravimetric and TDR values provide strong evidence of the general influence of *BWE* changes on *N*_0_. Estimates of specific *N*_0_ values will be somewhat influenced by which method or weighting scheme is used.

### 4.3. Vegetation Correction Factor

Developing the correction factor for biomass in row crops can overcome a critical issue for implementing CRNS in agricultural contexts. As discussed earlier, there were first- and second-order correction factors, *f_p_*, *f_i_*, and *f_v_*, for neutron intensity. We considered the correction factor *f_b_* a second- or third-order correction factor, further improving CRNS measurement accuracy. The values for *η* in the *f_b_* equation in this study ranged from 0.6 to 1.7%, with the middle at approximately 1%. The proposed correction factor for this study is that for every mm of *BWE*, there was an ~1% reduction in neutron intensity. This was determined by averaging all neutron intensity reduction values presented in this study. In the form of the equation presented by Baatz et al. [24] and Heistermann et al. [54], the *f_b_* equation can be written as seen in Equation (5). The average value of *η* from this study was 0.01067. With this value determined, the only input required in the equation is a value for *BWE*. With further confirmation of *η*, the *f_b_* correction can be integrated into the overall neutron correction equation as presented in Equation (4) above. Additional validation studies will likely be required before this correction factor is fully integrated into the above correction equation for other crops and forest systems.

Many prior studies have acknowledged the need for an *f_b_* correction value (Table 1). However, the value of *η* in *f_b_* has only been directly quantified three times [24,25,26]. The studies performed by Baatz et al. [24] and Vather et al. [25] agreed well with one another, considering the type of biomass sampling that occurred in each study. Both studies were performed in tree stands, primarily forested environments. However, Vather et al.’s [25] *f_b_* agrees more closely with the value presented here than Baatz et al.’s value. Baatz et al.’s study does not agree with the value presented here for row crops. The value presented here agrees well with Franz et al.’s [26] study, which concluded that there is a 1% decrease in *N*_0_ for every mm of *BWE* added. Franz et al.’s [26] study also occurred in a row crop environment in Nebraska. This study and that of Franz et al. [26] both conclude that a row crop *η* of 1% is suitable.

### 4.4. Variations in Sensor Type

Table 4 presents data regarding the two different sensor types used in this study. Recall that CSP1 and CSP2 used the CRS-2000/B, while CSP3 used the CRS-1000/B. The difference between these two sensors was primarily the volume of gas used and the difference in neutron counts. The CRS-2000/B had approximately double the size and double the neutron counts. Table 4 shows that the CRS-2000/B had a larger *f_b_* value than the CRS-1000/B sensor. This value was nearly doubled for the CRS-1000/B sensor compared to the CRS-2000/B sensor. Despite the difference in count rate and associated uncertainty levels, it is not immediately clear to the authors why the slope ratios may vary by a factor of 2. We do not believe this is due to the geometry of the CRNS detector design. Likewise, we do not think the result is due to the nature of vegetation in rainfed vs. irrigated areas, although rainfed crops will develop more extensive below-ground biomass due to water stress. We suggest further studies with the same detector with sufficient counting rates to eliminate this confounding effect.

### 4.5. Limitations of Study and Future Directions

The neutron intensity reduction factor, *η* (% reduction per mm *BWE*), is the critical metric for accounting for changes in vegetation. In the case of Baatz et al., *η* was 0.5%, but for this study, all values were higher than that, with an average of ~1%. The first potential reason for this difference is the place and time in which the studies occurred. The data presented here are from a long-term study in a crop environment, whereas Baatz et al. [24] conducted a 3-year study (with only 1 year of biomass measured) in a primarily forest environment. Trees and row crops have different growth cycles and internal water dynamics (Figure 2). *BWE* differs in quantity and degree of fluctuation for forests and row crops. Forests have large quantities of water in their biomass, and thus a larger *BWE*, and subtle seasonal fluctuations in water content. Conversely, row crops have smaller *BWE* values but rapid, sharp seasonal peaks and vast changes in water content. These variations create questions as to whether or not *η* and/or the *f_b_* equation may be different for forests and row crops. Another important difference to consider between this study and Baatz et al.’s [24] study is that while there were more CRNSs in the German forest, they only measured one year of biomass where there was a deforestation experiment. This allowed them to study the pre-cut environment, the environment immediately after cutting, and the early regrowth of the tree stand. While this did capture an extreme difference in biomass, it did not capture the same *BWE* variation in maize or soybeans.

Another key difference between row crop and forest environments is how their hydrogen pools are distributed in space. In crop environments, the hydrogen in *BWE* is assumed to be dispersed as a layer of water on the surface, whereas forests have clusters of hydrogen pools in their trunks. This difference in the location of hydrogen pools likely creates a geometric effect. Franz et al. [40], Andreasen et al. [19,30,55], and Kohli et al. [56] explored the influence of vegetation geometry on neutron transport simulations, which suggests that clumpy vegetation has a reduced effect on neutron count intensity as compared to a layer of vegetation with the same mass of hydrogen.

Another factor creating discrepancies between *f_b_* in this study and previous studies is data uncertainty. With a sparse number of TDR measurements of SWC, there is a large amount of variability when using it for CRNS studies. This variability propagates through Equation (2) to create noise in the resultant *N_pvi_* value. This noise is most evident in the analysis using the CRS-1000/B data.

An additional reason for the differences between Baatz et al.’s correction factor and this study is that we combined all remaining hydrogen pools here. This included the presence of other hydrogen pools in maize that cannot be easily accounted for. One key unaccounted hydrogen pool is the amount of water present on the plant itself as interception storage. Crops intercept water from fog, humidity, and rain, and this water settles on their surface. This interception can be considerable in maize (~0.5–2 mm *BWE*), as evidenced by walking through a field in the morning and becoming drenched. Given that there may be a strong diurnal cycle of this interception and evaporation process, future studies using larger neutron detectors may be able to quantify it directly. When the *BWE* in maize is at its peak of 7 mm, a 0.5–2 mm addition of intercepted water may be significant and detectable within the overall noise. By not considering this additional hydrogen pool and sampling occurring primarily in the morning hours, the *f_b_* values reported in this study may have been larger than those in past studies like that of Baatz et al. [24].

A final potential source of variation between Baatz et al.’s [24] study and this study is that there are microclimate effects within the plants in row crops. Within the maize canopy, there are extensive variations in the humidity and air temperature. The neutron sensors may not capture the impacts of the microclimate, as the environmental sensors on the CRNSs are often placed above or outside the maize and soybean canopies. The proximity of sensors to biomass may influence the neutron counts. The sensors closer to the biomass may experience some of the microclimate effects, while the sensors further away do not have this same influence. Microclimate effects can vary throughout the CRNS footprint, and they cannot be easily measured or differentiated by the CRNS. Microclimate influences may have caused the counts to be somewhat reduced but were not accounted for in the correction factors.

It is possible that the *f_b_* proposed in this study overcorrects for *BWE* in row crops, particularly due to the microclimate and unaccounted layers of water discussed above. Overcorrecting for biomass would create new sources of error and should be avoided. With the preliminary findings of Franz et al. [26], Vather et al. [25], and this long-term study, we now have three studies that support a 1% reduction in neutron intensity per unit *BWE*. Both the extensive gravimetric and TDR SWC datasets support a correction factor of around 1% for the larger detector types and in irrigated crops. The data from the smaller detector type (with significantly more noise) located in a rainfed crop supports a correction factor closer to 0.5% per unit *BWE*, which aligns with other studies collected in forests. To accurately determine if there is a difference between *f_b_* for crops and forests, a long-term study of *f_b_* in forested environments may be necessary. We also suggest a study investigating the diurnal cycle of intercepted water and microclimate variations in row crops. This would require a large detector, which presents additional logistical factors for use in row crops.

## 5. Conclusions

This work contributes a greater understanding of vegetation’s influence on neutron intensity measurements by providing a long-term study in three field sites in Eastern Nebraska with two types of CRNS. The primary goal of this study was to confirm the need for an *f_b_* factor and determine the neutron intensity reduction coefficient (*η)* for various row crops. Neutron counts, biomass, and SWC measurements were used for up to 13 years in a maize–soybean rotation to determine if a single *η* value can be used across all row crops. The first key finding was that a linear relationship existed between *N*_0_ and *BWE*. It was found that there was a 1% reduction in neutron intensity for every mm of *BWE* on the surface (*η* = 0.01). Possible *η* values ranged from 0.6% to 1.7%. These values agreed with two prior studies but doubled the commonly accepted *η* found for forest ecosystems. The framework proposed here can be applied to forests, but destructive sampling of *BWE* is challenging. The value of *η* for forests has been proposed to be ~0.5% by Baatz et al. [24], but a long-term validation study is needed.

This work also found that *η* varied between the two detector types (CRS-1000/B and 2000/B), potentially due to increased noise and below-ground biomass differences between irrigated and rainfed crops. The *η* value was based on both continuous sparse TDR networks and spatially exhaustive gravimetric surveys. The *η* determination in this study was specific to this site in row crop fields in Nebraska. Future long-term studies should be performed elsewhere to confirm these results, particularly in forest ecosystems or in cropping systems with significantly more below-ground biomass (i.e., sugar beets, potatoes, cassava, etc.). For forest systems, this would require methods to measure biomass through other metrics, as *BWE* and other destructive sampling techniques are not practical. The CRNS community continues to propose, test, and refine an acceptable vegetation correction factor. This study advances our understanding of row crops and will help push the CRNS community toward an acceptable framework to further improve the accuracy of CRNS SWC monitoring.

## Figures and Tables

**Figure 1 sensors-24-04094-f001:**
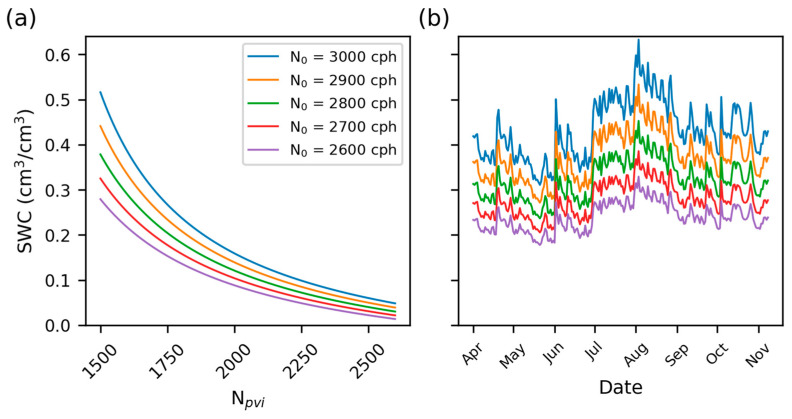
The variations in SWC estimates with varying *N*_0_ calibration parameter estimates for a typical CRNS count rate (*N_pvi_*). Panel (**a**) demonstrates that a neutron count measurement differing by an *N*_0_ of 100 counts will lead to an approximately 0.10 cm^3^ cm^−3^ change in resultant SWC on the water retention curve. Panel (**b**) shows the impact on SWC for a hypothetical growing season for a CRNS in Nebraska. Note that both panels share the y-axis, and the colors represent the same *N*_0_ values for each panel. Section 2.3 will provide more details on the calibration function.

**Figure 2 sensors-24-04094-f002:**
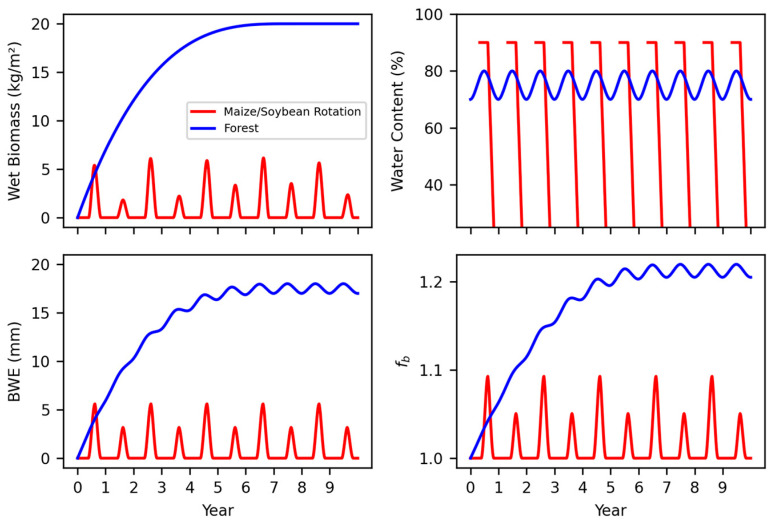
Conceptual figure capturing the annual changes in vegetation wet biomass (kg m^−2^), vegetation internal water content (%), *BWE* (mm), and the vegetation neutron correction factor (*f_b_*). The two types of biomass presented are a maize–soybean rotation and a forested environment. Note that the crop water content begins high and drops at senescence after reaching maturity. This figure was created using realistic values of wet biomass, water content, and BWE for both the maize–soybean rotation and forest environments. These values are only hypothetical, as this is a conceptual figure for demonstration purposes.

**Figure 3 sensors-24-04094-f003:**
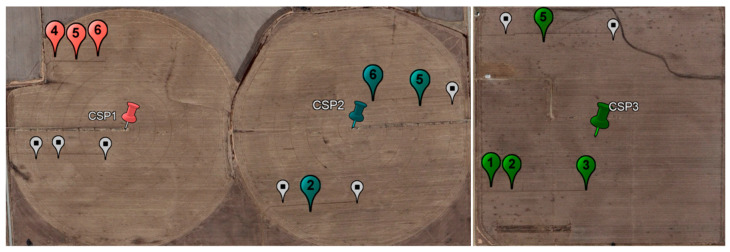
Locations of CRNS and profiles of in-situ soil water content (SWC) time domain reflectometry (TDR) sensors in this study for sites CSP1, CSP2, and CSP3. TDR sensor depths were 10, 25, 50, and 100 cm for all sites and 175 cm for CSP3 only. Field sites CSP1 and CSP2 had three profiles, and CSP3 had four. TDR sensors are denoted with location markers and their associated IMZ number (The IMZ numbers used are abbreviations of the Ameriflux network sensor names. In CSP1, IMZ4 is labeled SWC_F_1, IMZ5 is SWC_F_2, and IMZ6 is SWC_F_3. In CSP2, IMZ2 is SWC_F_1, IMZ5 is SWC_F_2, and IMZ6 is SWC_F_3. In CSP3, IMZ1 is SWC_F_1, IMZ2 is SWC_F_2, IMZ3 is SWC_F_3, and IMZ5 is SWC_F_4.), while CRNS sensors are denoted with thumbtacks. Gray markers show the IMZs where only vegetation samples were collected and no TDR sensors were present.

**Figure 4 sensors-24-04094-f004:**
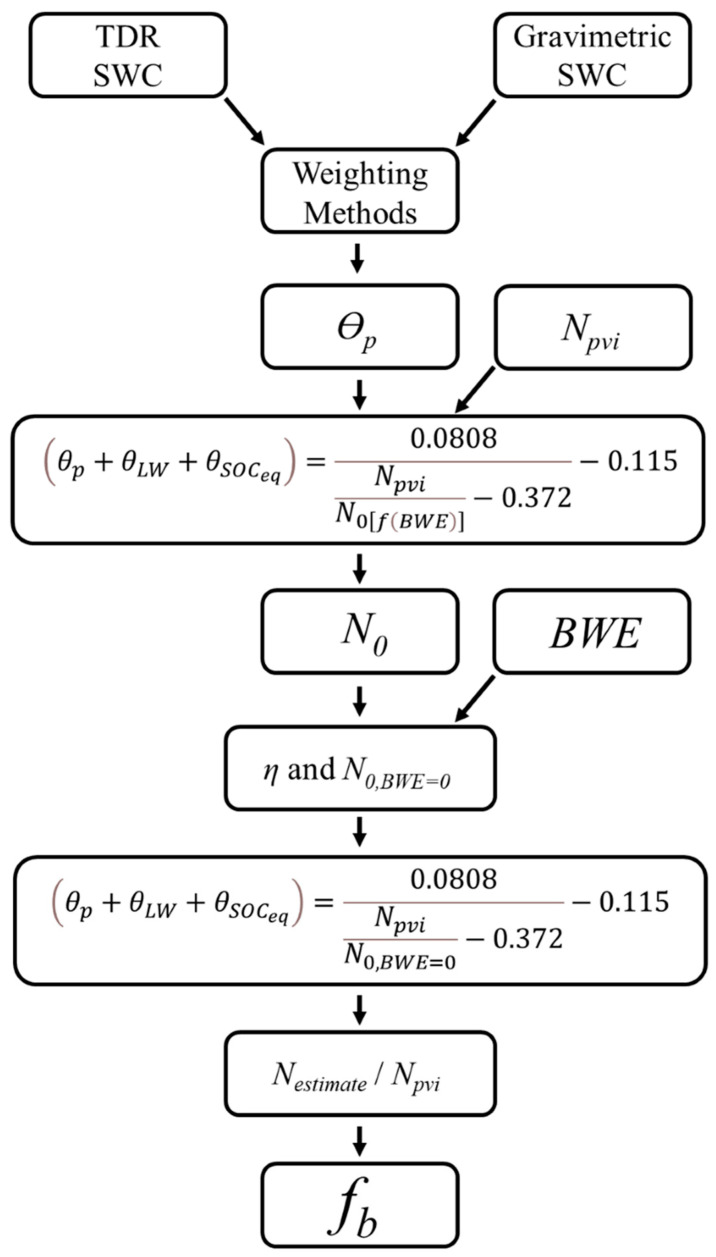
Flow diagram of calculations used in this study. Note that *η* is only a result of plotting *N*_0_ vs. *BWE* and is not used in Desilets et al.’s equation in the next step.

**Figure 5 sensors-24-04094-f005:**
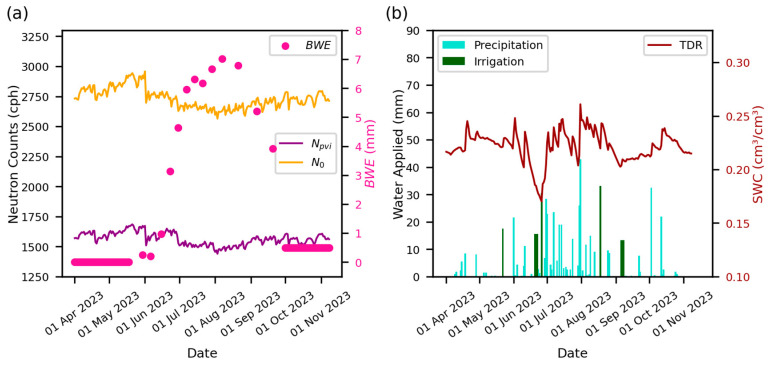
(**a**) Plot of *N_pvi_*, *N*_0_, *BWE* time series. (**b**) A bar chart of the irrigation and precipitation (mm) at site CSP1 in 2023 and the corresponding SWC time series. The same plots for all other years and fields can be found in the Appendix A. The TDR values were weighted with the 10 cm sensor having 75% contribution and the 25 cm sensor having 25% contribution. The averaged values of all TDR sensors in the field were used to plot the line.

**Figure 6 sensors-24-04094-f006:**
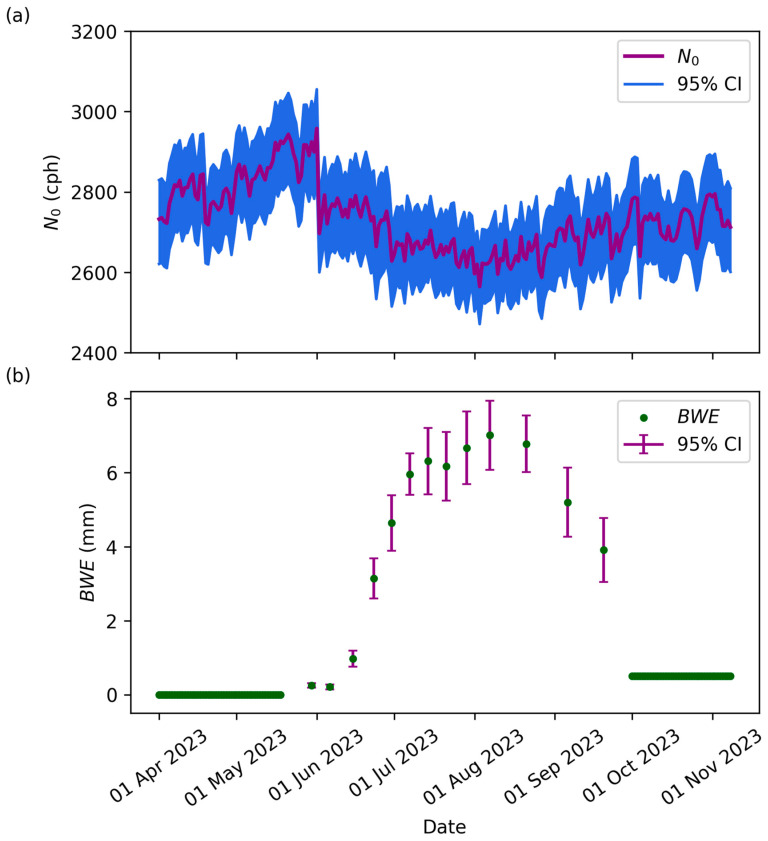
(**a**) Time series of *N*_0_ using TDR in Equation (4). Error bars represent the 95% confidence interval. For *N*_0_, the uncertainty from the TDR was propagated through Equation (4). (**b**) The corresponding *BWE* for CSP1 in 2023. The uncertainty for *BWE* was calculated from the six replicate plots across the field on each sample date. The *BWE* uncertainties were calculated using standard errors.

**Figure 7 sensors-24-04094-f007:**
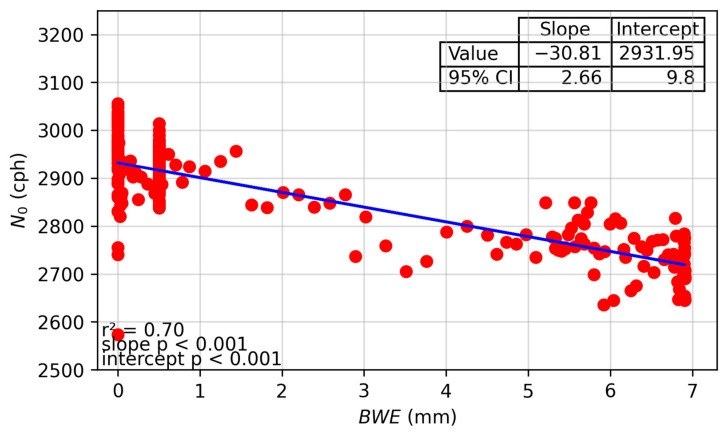
Scatterplot of *N*_0_ vs. *BWE* for CSP1 for 2023. This plot presents the slope and intercept values with their corresponding 95% confidence intervals, *p*-values, and the coefficient of determination for the relationship. Residual values are likely caused by noise in TDR measurements and their spatial averaging with a small number of sensors.

**Figure 8 sensors-24-04094-f008:**
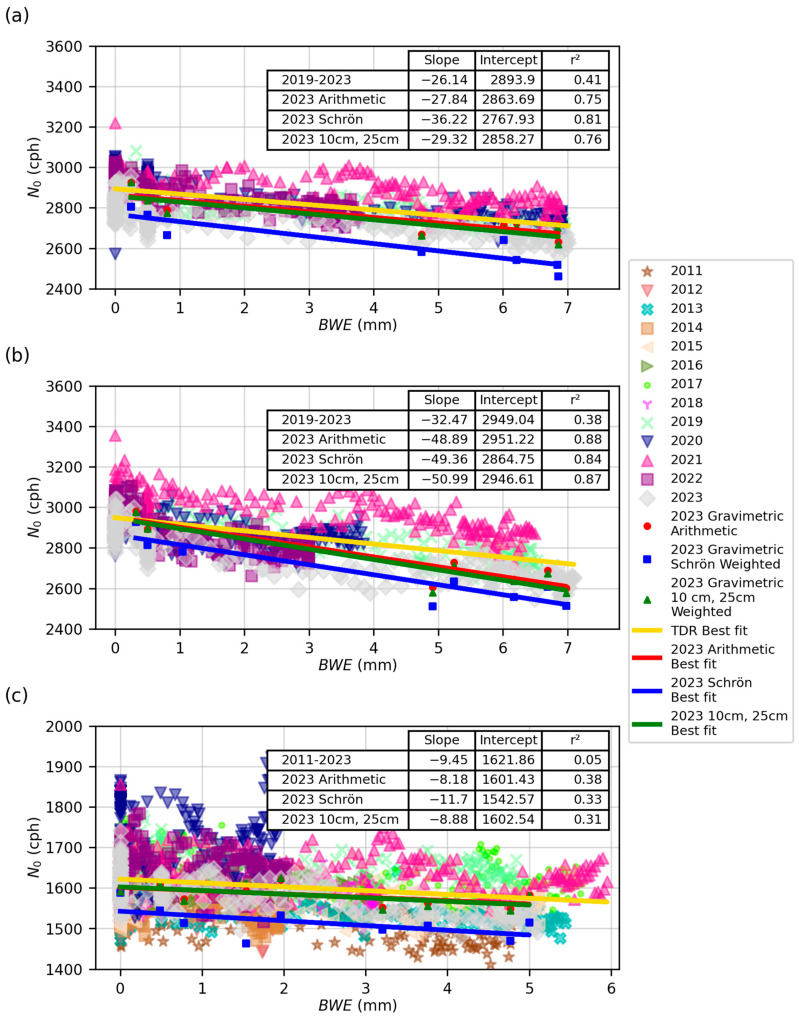
Relationship between *N*_0_ and *BWE* for all sites and all years using the TDR and gravimetric samples using the three weighting schemes. (**a**) CSP1, (**b**) CSP2, and (**c**) CSP3.

**Figure 9 sensors-24-04094-f009:**
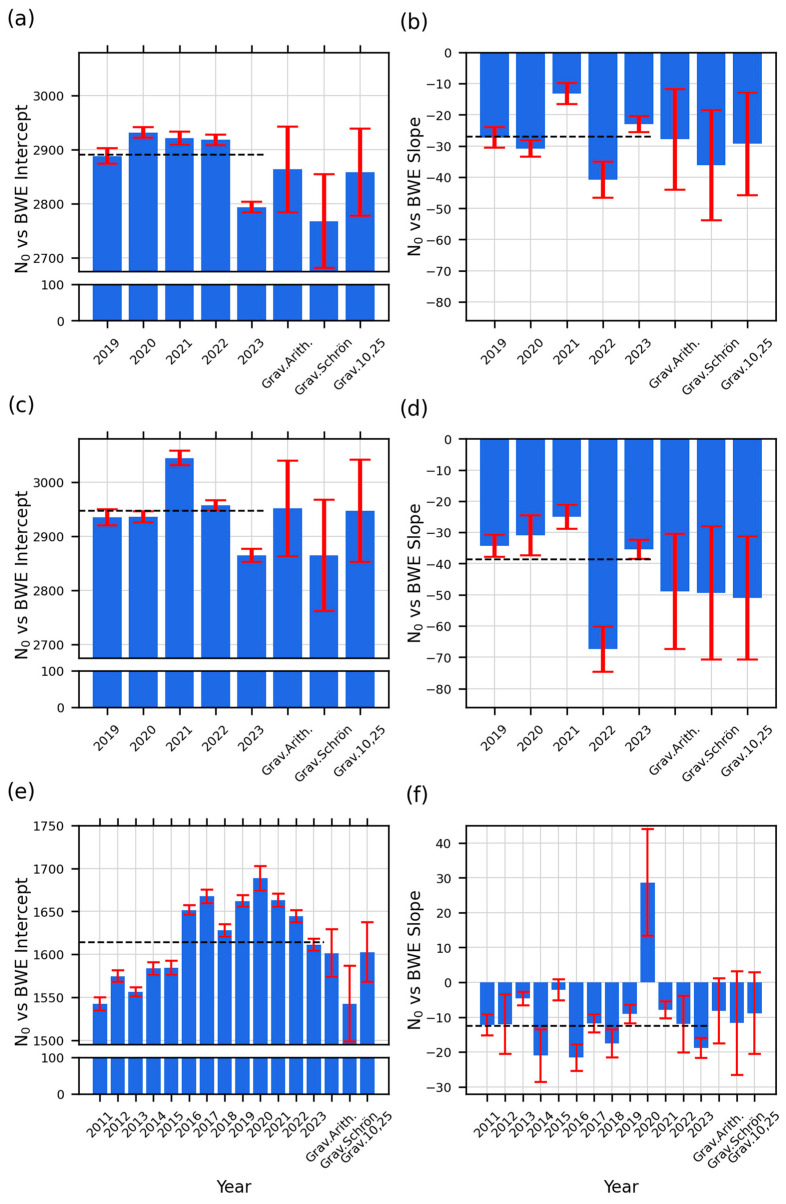
Yearly plot of *N*_0_ vs. *BWE* slopes and intercepts for (**a**,**b**) CSP1, (**c**,**d**) CSP2, and (**e**,**f**) CSP3. (**a**,**c**,**e**) are intercepts, and (**b**,**d**,**f**) are slopes. Please note the broken axis on the intercept plots. Error bars show the 95% confidence intervals. The black dashed lines denote the overall average of TDR slopes and intercepts within each site.

**Figure 10 sensors-24-04094-f010:**
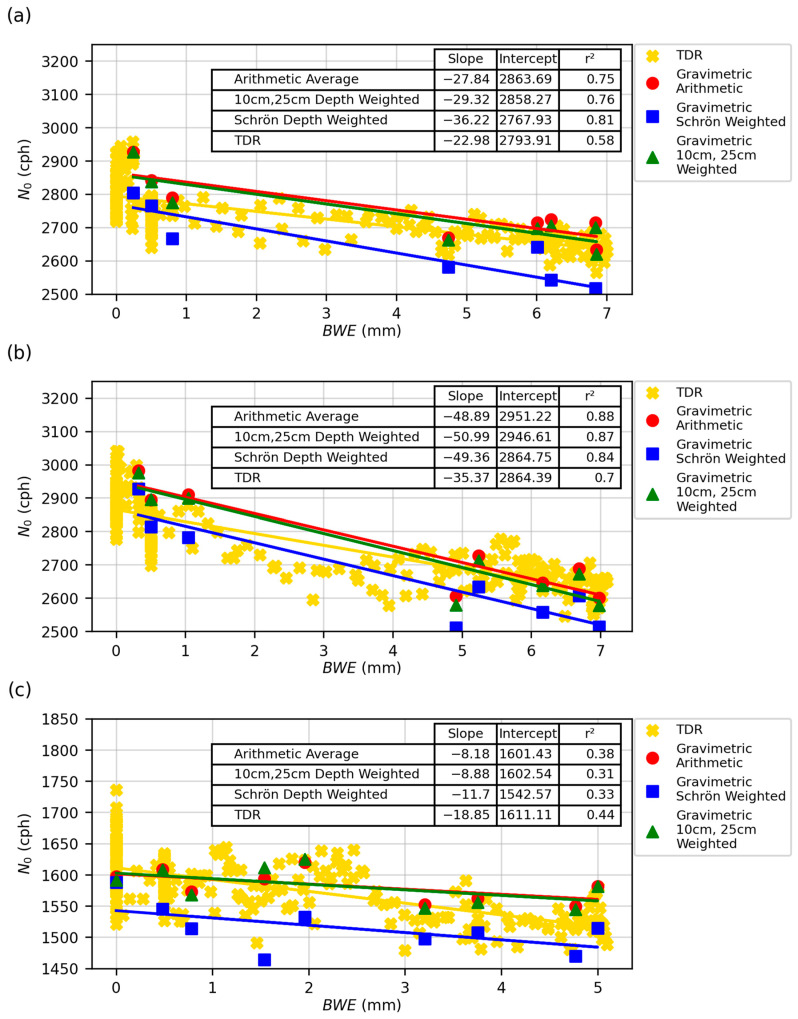
Plots of *N*_0_ vs. *BWE* for both TDR and three gravimetric weighting methods for only the year 2023 for fields (**a**) CSP1, (**b**) CSP2, and (**c**) CSP3.

**Figure 11 sensors-24-04094-f011:**
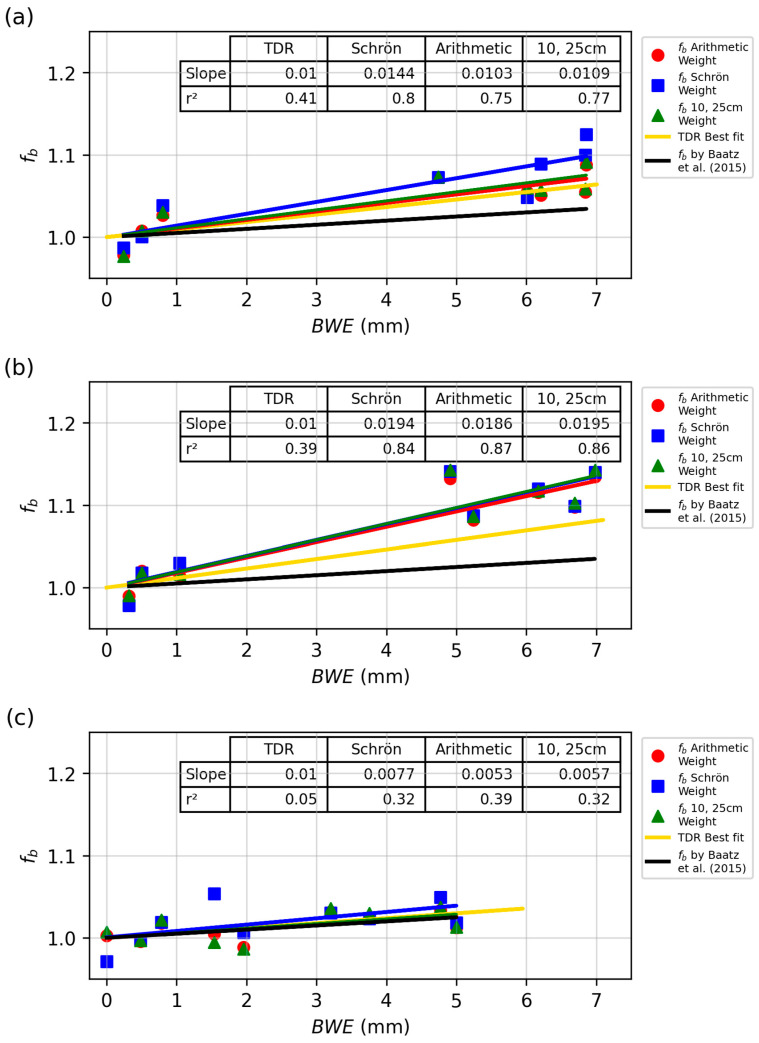
Relationship between *f_b_* and *BWE* for all sites and all years using the TDR and gravimetric samples with three weighting schemes. (**a**) CSP1, (**b**) CSP2, (**c**) CSP3. Note the black line shows the proposed value by [24].

**Table 1 sensors-24-04094-t001:** Summary of CRNS methods used to estimate biomass influence on CRNS measurements. The T/E is the thermal/epithermal method, *BWE* is biomass water equivalent, and AGB is above-ground biomass. An expanded version of this table with specific study results can be found in the Appendix A.

Method	Method Summary	Pro	Con	*f_b_*	Source
Thermal/ Epithermal Ratio	Uses bare and moderated sensors to determine the ratio between thermal and epithermal neutrons	Minimizes need for soil sampling, easy to measure, many applications	Requires two detectors, limited validation, several assumptions	Biomass presence reduces *N*_0_ by 13.8% for 13.7 kg/m^2^ dry biomass [25] linear relationship between ratio and biomass present	[19,25,28,29,30,31]
*BWE*	Weigh plants wet, oven dry, weigh plants dry, remove cellulose signal	Accurate, accounts for all parts of plants, used for ground truthing of other methods	Destructive, time-consuming, poor temporal resolution, one sample may not represent field	Linear relationship can be used to correct for vegetation, 1% decrease in *N*_0_ per mm *BWE* added	[26]
Remote Sensing	Satellite measurements taken to estimate biomass	Generally inexpensive, easily measured, large-scale coverage, non-destructive	Poor spatial and temporal resolution, atmospheric interference, complex analysis, lack of ground validation	No direct values reported, many studies discuss potential for an *f_b_*	[32,33,34,35]
Above-Ground Biomass	Measurements of AGB through several methods (destructive, allometric, etc.)	Accurate, reasonable biomass estimates, cost effective	Does not account for belowground biomass, labor-intensive, does not capture field variability	One study uses [24] correction, most studies report linear *N*_0_ and AGB relationship, no direct *f_b_* values reported, but need is addressed	[22,27,36,37,38]
Plant Allometry	Uses easily measured plant parts to estimate biomass of whole plant	Easy to measure and allows predictive power in growing biomass	Does not capture biomass variations and field variability, complex calculations	Biomass relationship listed as linear in some studies and non-linear in others, no direct *f_b_* reported	[39,40,41]
Scaling	Linear scaling approach used to upscale soil moisture measurements	Low cost, can be used at different time intervals (daily vs. seasonal trend), easy to upscale or downscale	Requires point water measurements, needs *N*_0_ calibration, relies heavily on z* and vertical weighting	No *f_b_* reported, reported that fast-growing biomass (e.g., maize) adds ~7 mm of *BWE*	[42]
Combination of Methods	Uses at least two methods listed above	Allows cross validation, more comprehensive study, and limits measurement bias	Adds complexity, requires more resources, introduces potential error sources	0.9% per kg/m^2^ of dry AGB or per 2 kg/m^2^ of *BWE* [22], Other studies report an *f_b_* is needed, but do not quantify	[20,24,43]

**Table 2 sensors-24-04094-t002:** Laboratory and sampling results for $\theta_{LW}$ (g g^−1^), $\theta_{{SOC}_{eq}}$ (g g^−1^), and bulk density ($\rho_{b}$) (g cm^−3^) utilized in Equations (2) and (4). Note that CSP3 was sampled in 2011 and 2023. Shown with the $\rho_{b}$ values are 95% confidence intervals, which were calculated using the standard deviations.

	2023 CSP1	2023 CSP2	2023 CSP3	2011 CSP3	Value Used
$\theta_{LW}$	0.055	0.059	0.06	0.0375	0.058
$\theta_{SOC}$	0.0113	0.0067	0.0097	0.006	0.0092
$\rho_{b}$	1.40 ± 0.25	1.42 ± 0.17	1.43 ± 0.38	1.42 ± 0.17	1.42

**Table 3 sensors-24-04094-t003:** Summary of slope, intercept, *η* for TDR, and 2023 gravimetric data by year and site. The error presented represents the 95% confidence intervals. Please note that 2020 in CSP3 is not included in the TDR averages.

Year	CSP1 Intercept	CSP1 Slope	CSP2 Intercept	CSP2 Slope	CSP3 Intercept	CSP3 Slope	CSP1 *η*	CSP2 *η*	CSP3 *η*
2023 Grav. Arithmetic	2863.69 ± 79.0	−27.84 ± 16.1	2951.22 ± 88.5	−48.89 ± 18.4	1601.43 ± 27.6	−8.18 ± 9.3	−0.97	−1.66	−0.51
2023 Grav. Schrön et al.	2767.93 ± 86.5	−36.22 ± 17.7	2864.75 ± 102.7	−49.36 ± 21.4	1542.57 ± 44.1	−11.7 ± 44.1	−1.31	−1.72	−0.76
2023 Grav. 10, 25 cm Wt.	2858.27 ± 80.4	−29.32 ± 16.4	2946.61 ± 94.6	−50.99 ± 19.7	1602.54 ± 34.8	−8.88 ± 11.7	−1.03	−1.73	−0.55
2023	2793.91 ± 9.9	−22.98 ± 2.6	2864.39 ± 11.8	−35.37 ± 3.1	1611.11 ± 7.0	−18.85 ± 2.8	−0.82	−1.23	−1.17
2022	2918.41 ± 9.7	−40.84 ± 5.7	2957.0 ± 9.6	−67.36 ± 7.2	1644.22 ± 7.0	−11.95 ± 8.1	−1.40	−2.28	−0.73
2021	2920.98 ± 11.9	−13.28 ± 3.4	3044.36 ± 13.5	−24.99 ± 3.9	1663.1 ± 7.5	−7.91 ± 2.4	−0.45	−0.82	−0.48
2020	2931.95 ± 9.8	−30.81 ± 2.7	2935.68 ± 10.5	−30.98 ± 6.4	1688.51 ± 14.3	28.64 ± 15.3	−1.05	−1.06	1.70
2019	2888.29 ± 14.1	−27.27 ± 3.4	2935.06 ± 14.7	−34.32 ± 3.5	1661.97 ± 6.7	−9.12 ± 2.6	−0.94	−1.17	−0.55
2018					1627.93 ± 7.2	−17.5 ± 4.0			−1.07
2017					1667.4 ± 8.0	−11.79 ± 2.6			−0.71
2016					1651.59 ± 5.7	−21.61 ± 3.8			−1.31
2015					1584.47 ± 8.1	−2.11 ± 3.0			−0.13
2014					1583.61 ± 7.4	−21.05 ± 7.6			−1.33
2013					1556.41 ± 5.1	−4.67 ± 1.9			−0.30
2012					1574.7 ± 6.7	−12.04 ± 8.6			−0.76
2011					1542.13 ± 7.6	−12.16 ± 3.0			−0.79
TDR Avg.	2890.7 ± 11.1	−27.0 ± 3.5	2947.3 ± 12.0	−38.6 ± 4.8	1614.1 ± 7.0	−12.6 ± 4.2	−0.9 ± 0.3	−1.3 ± 0.4	−0.59 ± 0.4

**Table 4 sensors-24-04094-t004:** Values of *f_b_* (neutron intensity reduction) and the slope ratio for all gravimetric methods and TDR measurements for all three sites. Sensor-specific, gravimetric, TDR, and overall averages are also presented.

Field and Weighting Method	Neutron Intensity Reduction (% per mm *BWE*)
CSP1 Schrön et al.	1.309
CSP2 Schrön et al.	1.723
CSP3 Schrön et al.	0.758
CSP1 Arithmetic	0.972
CSP2 Arithmetic	1.639
CSP3 Arithmetic	0.511
CSP1 10, 25 cm	1.026
CSP2 10, 25 cm	1.730
CSP3 10, 25 cm	0.554
CSP1 TDR	0.903
CSP2 TDR	1.101
CSP3 TDR	0.583
CRS-2000/B Average	1.300 ± 0.24
CRS-1000/B Average	0.602 ± 0.11
Gravimetric Average	1.136 ± 0.32
TDR Average	0.862 ± 0.30
Overall Average	1.067 ± 0.25

## Data Availability

The data presented in this study are freely and openly available on Appendix A.

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
