# Peer review of "Effect of Biomass Water Dynamics in Cosmic-Ray Neutron Sensor Observations: A Long-Term Analysis of Maize–Soybean Rotation in Nebraska"

_sensors, 2024, doi:10.3390/s24134094_

Round 1

Reviewer 1 Report

Comments and Suggestions for Authors

I have had the opportunity to review your article titled "Precise Soil Water Content Measurement for Effective Water Resource Management" with great interest. I commend you for conducting such a comprehensive study that addresses a crucial aspect of water resource management, particularly in agricultural settings. Overall, I find the research to be well-conducted and insightful, providing valuable contributions to the field. However, I have identified a few areas where minor revisions could further enhance the clarity and impact of your work.

1. Clarification on the Introduction section

The introduction section does an excellent job of highlighting the significance of the state of the art, and the relationship with your findings, particularly regarding the vegetation correction factor for CRNS.But it would be necessary to improve the Fig.1 explanation. How it is created?How/where is it taken from? Readers should be able to understand the significance of the figure without having to refer back and forth between the main text and the literature.

2. Material and Methods Section

- How do the two CRNS sensors vary? Is it in terms of their geometry or the type of detector used?" (line 146) to "Could you elaborate more on the main characteristics that distinguish the two CRNS sensors? For example, their geometry and the type of detector employed?" (line 250)

- Table 2 does not include any uncertainties in the reported results. Are these uncertainties calculated? In the text, it is mentioned that they are derived from laboratory tests. Could you provide further clarification on how these values and uncertainties are determined from the laboratory tests?

- The inclusion of Fig. 3 in the main body of the article is not essential; too large, anyhow. However, it could be included as supplementary material for the article.

- Figure 5: Were the uncertainties in the BWE calculated using standard deviations?

-In Figure 6, all uncertainties seem to have disappeared. What is the reason for this? Is the statistical fit accurate?

Fig. 7: What are the acceptable ranges for variations in intercept and slope? Both appear to fluctuate, yet in the text, it was mentioned that the slope is not influenced by the three weighting schemes.

  Fig.8: the slopes and intercepts are not statistical consistent in total, for example in plot c the first two are consistent, but not with thethird ot fifth. In plot e it is more evident, from 2016 the results are not more consistent with the results before.Can you in fact provide additional clarification or context regarding the comment on lines 374-375 on this topic?

Table 3: due to the intrinsic differences between the crns detectors, and the different time tests, what is the main results from table 3? can you comment further about the sentence "414, despite the uncertainty..they are consistent"?

Fig.9 the same comments as before. (no uncertainty, no comments on the slope and intercept are presented)

Comments on section Discussion

Can you further comments on line 503 -510, which is a relatively known problem?

Reviewer 2 Report

Comments and Suggestions for Authors

This study gave us a better understanding of vegetation’s influence on neutron intensity for row crops. The database is solid and the introduction and discussion were appropriately addressed. I understand that this study contains multiple years of data, however, the results section needs to be shortened. The only curious question is why the SWC results of CRNS in three field sits were not displayed. Some minor questions are as follows:  

Line 2: Effect of … on…

Line 184: was the comparison only done in 10 and 25-cm depths? How about the other three depths?

Line 186: is there any supports or references for weightings of 75% and 25%?

Line 188: how to understand the increased influence of the upper depth?

Line 196: delete the sentence of what has been reported in the literature.

Line 230: change 70 deg. C into 70 ℃.

Reviewer 3 Report

Comments and Suggestions for Authors

Review of manuscript sensors-3003165 Effect of biomass water dynamics in cosmic-ray neutron sensor observations: A long-term analysis of maize-soybean rotation in Nebraska by Morris et al.

Summary

The Authors present a study about biomass correction for improving soil water content estimation by means of cosmic ray neutron sensing techniques. The topic is relevant. The Authors present a very good data set and a scientifically sounded analysis. In general, the manuscript is well structured and written but some figures are a bit redundant. Discussion is reached. In suggesting possible improvements, I have listed below one major change in the presentation of the results that might improve the clarity of the manuscript and several but minor specific comments.

Major comments

I appreciated the decision to show one year and one location as example (Section 3.1) and then the overall results (section 3.2 and Figure 7). The year selected is however the 2020 where only TDR observations are available. The comparison to the gravimetric is shown later in section 3.3. I suggest the Authors showing first the results of the year where you have also the gravimetric samples and discussing the differences between using TDR and using the gravimetric. This would help understanding the results and would strengthen the use of TDR over all the years.

Specific comments (L – line number)

L63. 3rd instead of 2nd?

L160 Any comments on the plant residues? They were left in place for soil cover. If too dry to affect BWE and fb factor, they might affect soil carbon?

L183. CRNS is sensitive to the upper soil layers but TDR have been installed at 10 cm and down. Any comments? A discussion when comparing TDR and gravimetric could be reported not only in the discussion section but also when results are presented.

L196. I have seen some effort of the Authors in comparing the weight functions. Is still debated if one or other is better? Please elaborate a bit on that on the manuscript to help the readers to understand the motivation behind.

L220. I would add a short description about how LW and SOC were determined or add a reference where it is described. Since you work with long-term experimental sites, any idea about the changes of these values in time? Also in relation to the plant residues?

L229. Adding the IMZ area and the plant sampling location at Figure 2 would help the reading.

L279. I do not really see a similar general patterns? Why should that be? Please elaborate on that.

Figure 4b shows a remarkable drop in TDR observations in April. Can this be explained or rather a sensor malfunction? This dynamic also affects the N0 in figure 4a but might be considered an error. I suggest a quality check of the TDR data before looking at N0 and BWE relation. CRNS SWC could also be added and discussed in Figure 4b. Please also clarify if this TDR time series is the average of all the sensors (location and depth) within the field.

Figure 5. Am I wrong or these plots show the same as figure 4a but with bands? I think the figures are redundant and I suggest to rather keep only one, e.g., to show in figure 4b the time series as in figure 5.

In Figure 5 I see that after harvesting BWE remains higher than zero, but I did not find information about it in the manuscript. Please clarify.

L356 – 361. Hard to read and follow this text. Better to summarize the message rather than writing all the years

L380. Does the seasonal precipitation can really explain the intercept variation? Why? Please elaborate on that.

L381. At CSP3, there seems to be a systematic shift to higher intercept values from 2016 to 2023. Could you explain that? Anything about the time changes of organic carbon? See comment above.

L436. Instead of “gravimetric data set” is SWC data set.
